# Evidence for the effectiveness of interventions to reduce mental health related stigma in the workplace: a systematic review

Mónika Ditta Tóth ![ORCID] ,[1] Sarah Ihionvien,[1] Caleb Leduc ![ORCID] ,[2,3] Birgit Aust ![ORCID] ,[4] Benedikt L Amann ![ORCID] ,[5,6] Johanna Cresswell-Smith ![ORCID] ,[7] Hanna Reich ![ORCID] ,[8,9] Grace Cully ![ORCID] ,[2,3] Sarita Sanches ![ORCID] ,[10,11] Naim Fanaj ![ORCID] ,[12] Gentiana Qirjako,[13] Fotini Tsantila ![ORCID] ,[14] Victoria Ross ![ORCID] ,[15] Sharna Mathieu ![ORCID] ,[15] Arlinda Cerga Pashoja ![ORCID] ,[16] Ella Arensman ![ORCID] ,[2,3,15] György Purebl ![ORCID] ,[1] MENTUPP Consortium

For numbered affiliations see end of article.

**Correspondence to**
Dr Mónika Ditta Tóth;
tmonika85@gmail.com

## ABSTRACT

**Objectives** Increasing access to mental health support is a key factor for treating mental disorders, however, important barriers complicate help-seeking, among them, mental health related stigma being most prominent. We aimed to systematically review the current evidence for interventions focusing on reducing stigma related to mental health problems in small and medium enterprises (SMEs).

**Design** Systematic review with a focus on interventions targeting mental health related stigma in the workplace in accordance with PRISMA guidelines. The methodological quality of included articles was assessed using the Quality Assessment Tool for Quantitative Studies Scale.

**Data sources** PubMed, Ovid Medline, PsycINFO, Scopus, and Cochrane databases and Google Scholar were searched from January 2010 until November 2022.

**Eligibility criteria for selecting studies** We included experimental or quasi-experimental studies about workplace interventions aiming to reduce stigma, where the outcomes were measured in terms of stigmatisation against depression, anxiety and/or other mental health problems.

**Data extraction and synthesis** Records were screened by two independent reviewers after inspecting titles and abstracts and a full-text read of the articles to assess whether they meet inclusion criteria. The results were synthesised narratively.

**Results** We identified 22 intervention studies, 3 with high quality, 13 with moderate quality and 6 with weak quality. Only 2 studies included SMEs, but no study focused on SMEs exclusively . The mode of delivery of the intervention was face to face in 15 studies, online in 4 studies and mixed in 3 studies. We found a significant reduction in stigmatising attitudes in almost all studies (20/22), using 10 different instruments/scales. Effects seemed to be independent of company size. Online interventions were found to be shorter, but seemed to be as effective as face-to-face interventions.

**Conclusions** Although we did not find interventions focusing exclusively on SMEs, it is likely that antistigma interventions also will work in smaller workplaces.

**Trial registration** PROSPERO: ID: CRD42020191307

## STRENGTHS AND LIMITATIONS OF THIS STUDY

⇒ The present systematic review was based on a comprehensive search identifying 22 studies providing an important update since a similar review published in 2016.

⇒ The methodological quality of the identified studies was assessed by two independent reviewers using the Quality Assessment Tool for Quantitative Studies Scale.

⇒ Given the diverse study designs and outcome measures, it was not possible to conduct a meta-analysis.

⇒ Only studies with quantitative measurement were included in this review, however qualitative studies could provide important additional information, especially about the mechanisms leading to changes in stigma attitudes.

⇒ The different types of stigma-related changes — knowledge, beliefs and behavior — could not be defined because of the search strategy and inclusion criteria.s.

## INTRODUCTION

Mental disorders can have significant consequences, not only on the individual level, but also on a societal and economic level. In the context of the workplace,[1 2] poor mental health has been linked with absenteeism and presenteeism[3–5] leading to decreased workplace performance, productivity and increased risk of unemployment.[6 7] Depression and anxiety are the two most common mental disorders globally, and are therefore also most likely to impact work performance and productivity.[8]

Increasing access to mental health support is a key factor for treating mental disorders. Research highlights several important barriers which complicate help-seeking, with mental

health related stigma being the most prominent.[9] Stigma can be defined as the convergence of several interrelated components, such as labelling, stereotyping, separation, status loss and discrimination which occur together.[10] This includes perceived stigma (also known as social stigma) relating to an individual's perception of what others think and feel, and personal stigma (also known as self-stigma) reflecting individual thoughts and attitudes restricting openness about mental health difficulties, increasing risk of social exclusion and limiting help-seeking behaviour.[11 12] In a nationwide US study, over 90% of first responders found stigma as a main barrier to seeking help for themselves.[9] International evidence indicates that experiences of stigma and discrimination lead to decreased use of mental-health related interventions, including workplace-based mental health promotion programmes.[13–15] Mental health related stigma can also lead to the breakdown of social connections including avoidance, rejection and a perception of reduced competence.[16] As a consequence, the person involved may experience lack of career development, reduction of responsibilities, inequity in workplace policies, and exclusion from work integration and social activities. Stigma has also been found to increase the risk of unemployment, job uncertainty, and reduce the likelihood of being hired.[17]

Addressing mental health related stigma is a central component of LaMontagne's[18] model for workplace mental health, which integrates preventing harm and reducing risk factors, promoting the positive aspects of work, and management of mental illness. Investing in mental health in the workplace via mental health promotion actions can not only improve mental health on an individual level, but also increase economic productivity.[19–21] Several workplace-based mental health promotion programmes have been implemented in the European Union, with the majority of these being conducted in large companies. This means that interventions are only reaching a small proportion of all employees as the majority (99%) of European Union based workplaces represent small and medium enterprises (SMEs).[22] Despite proportionally more people being employed by SMEs in comparison to larger companies, SMEs often lack the financial and/or human resources support for mental health promotion. Although face-to-face interventions seem to be more effective, research shows that online interventions can be time-effective and cost-effective, and also easily implementable which can be favourable for small enterprises with presumably limited budgets to implement mental health promotion activities.[23]

Although research has shown that stigma can lead to a number of negative consequences and is a barrier for workplace mental health promotion, more insight is required into how best to reduce stigma. A number of intervention studies investigating the effects of antistigma initiatives have been conducted during the last 10–20 years, and so far only one systematic review has been published.[24] This review identified 16 intervention studies targeting stigma of mental illness at the workplace. The review included research published between 2004 and 2014 and found support for antistigma interventions leading to improved employee knowledge and supportive behaviour towards people with mental health problems. They concluded that while the majority of interventions demonstrated a positive effect on employees' attitudes, there remained significant need for improved methodological quality in future evaluations. Specifically, selection bias might have contributed to the positive effects. In particular, one of the main findings indicated that the majority of the interventions were conducted with more highly educated supervisors or in job groups, with more highly educated employees, and in the public sector. This reduces the generalisability to most workplaces in other diverse sectors with less educated workers. Consistent with workplace mental health research in general, most of these studies were also conducted in larger organisations, and therefore not providing any knowledge about interventions designed to reduce stigma in SMEs. The currently ongoing intervention project Mental Health Promotion and Intervention in Occupational Settings (MENTUPP Project) aims to contribute to knowledge in this area. A comprehensive online intervention has been developed and is currently being tested in a number of SMEs across European countries and Australia.[25] This review has been conducted as part of the MENTUPP Project to enhance its evidence base.

Therefore, the main aim of this paper was to systematically review the current evidence for interventions focusing on reducing stigma related to mental health problems in SMEs in various sectors. A secondary aim of the review was to investigate the mode of delivery and intensity/duration of interventions.

## METHODS

### Review procedure

A systematic literature search was conducted with a focus on interventions targeting mental health related stigma in the workplace. The review was conducted in accordance with the PRISMA guideline process.[26] Peer-reviewed articles about workplace-based antistigma interventions were searched from January 2010 until 14 July 2021 via PubMed, Ovid Medline, PsycINFO, Scopus and Cochrane databases. An additional Google Scholar search was conducted. All results from the database search were uploaded to Covidence (www.covidence.org), an online tool for managing and streamlining systematic reviews.

### Study selection

The systematic review was conducted addressing the following inclusion criteria: (1) The sample included employees and/or owners/managers; (2) The intervention at the workplace was aimed to reduce stigma; (3) The outcomes were measured in terms of stigmatisation against depression, anxiety and/or other mental health problems; (4) Studies had an experimental or quasi-experimental design (including quantitative data); (5) The studies were published in English; (6) The intervention was delivered through the workplace; and (7) The studies were published between January 2010 and July 2021.

Studies were excluded based on the following criteria: (1) No evaluation of the intervention; (2) Only qualitative evaluation (eg, interview or focus group); or (3) No direct measure on stigma (studies with indirect measures of stigma, such as knowledge of mental health, or attitudes towards mentally ill patients, were excluded).

After duplicates were removed, the records were screened by two independent reviewers (GP, SI) following a two-stage procedure: (1) Inspecting titles and abstracts of the studies, and (2) A full-text read of the articles to assess whether they met inclusion criteria. In the case of disagreement, a consensus was made together with a third researcher (MDT; first author of the study).

### Search strategy
The search string was developed by GP and MDT, reviewed by SI and CL, and subsequently reviewed by a subject librarian at Semmelweis University, Hungary (see search keywords in online supplemental appendix 1). Terms related to the following themes were used: mental health related terms AND workplace related terms AND stigma-related terms AND intervention related terms.

### Included studies
Online supplemental figure 1 displays the PRISMA flow diagram which shows the decision points during the screening process.

The PubMed, Ovid Medline, PsycINFO, Scopus and Cochrane databases and Google Scholar were searched resulting in initial identification of 3479 articles. After removal of duplicates (n=221) title screening and abstract review was conducted for 3258 articles, of which 154 were retained for full-text screening, and 23 met criteria for inclusion. However two articles Reavley 2018 and 2021 reported about the same intervention study, which means that 22 intervention studies were identified.

### Data extraction
Data extraction by two coauthors for the articles after full-text review included the following and was independently crosschecked by a third reviewer (MDT): (1) Author and year; (2) Study design; (3) Number of participants at baseline and follow-up; (4) Gender of participants (5) Target group (6) Sector and size of organisation (7) Intervention; (8) Intervention intensity; (9) Country; (10) (online supplemental table 1) outcome measure on stigma; (11) Evaluation timepoints; (12) Main findings (online supplemental table 2).

The review was conducted according to PRISMA (Preferred Reporting Items for Systematic Reviews and Meta-Analyses) guidelines.[26]

### Quality assessment
The methodological quality of each included article was assessed using the Quality Assessment Tool for Quantitative Studies (QATQS) Scale,[27] based on the following aspects rated from weak to strong: selection bias, design, confounders, blinding, data collection method and dropout. The global rating was high in case of 'no weak rating', moderate in case of 'one weak rating' and weak in case of 'two or more weak ratings'. Quality assessment was finalised after two independent reviews by the first and second authors of this review, followed by a consensus meeting together with a third independent reviewer GP.

### Patient and public involvement
No patient was involved.

## RESULTS
### Study characteristics
Of the 22 included intervention studies, 7 were conducted in Canada, 6 in Australia, 4 in Great Britain, 2 in Germany, and 1 each in Sweden, Spain and Japan. Nine studies used a randomised controlled trial (RCT) study design and the remaining 13 used a quasi-experimental design. An overview of the studies is presented in online supplemental tables 1 and 2.

### Sector and size of organisation
A total of 22 interventions were used by the included studies, most of which (12/22) were conducted in public sector organisations, or in a mixture of public and private sector workplaces (4/22). Only four studies focused solely on private sector companies, and no sector-specific information was provided in two of the studies. The interventions enrolled different professional groups in varying positions including healthcare workers (2 studies), first responders (4), public servants (2), maintenance staff (2), governmental employees (2), housing association (1), managers, leaders (8), hospitality industry (1).

Six studies provided information on the size of the organisations, the four studies in the private sector enrolled large enterprises with more than 250 employees. Two interventions enrolled a mixture of small, medium and large organisations. No intervention study specifically focused on SMEs.

### Quality assessment of the studies
The assessed methodological quality of the included studies varied from weak to strong, with three considered to be of high quality. Almost two-thirds of papers (13/22) were assessed as having moderate quality, most lacking a control group design. Six articles were appraised as weak, a rating driven primarily from low agreement rate and/or high dropout rate (online supplemental table 3).

The detailed evaluation criteria of the QATQS Scale are presented in online supplemental table 4.

### Interventions
Overall, 10 interventions used previously developed standardised interventions, including the Mental Health First Aid programme, Psychological First Aid, Applied Suicide Intervention Skills Training, Beyond Blue or Mental Health Guru, with other interventions being designed or modified to fit a workplace-based context. Twelve interventions used non-standardised mental health approaches. In terms of implementation, 4 interventions

included in the studies were delivered online, 15 delivered in person and three were blended interventions (delivered both online and face to face). All programmes used multimodal approaches, which included multiple intervention techniques such as psychoeducation, interactive skills training exercises and case vignettes/videos of experts with lived experience. Some of the interventions contained specific leadership-focused elements. The most frequent topics were: education about the features and symptoms of mental disorders (special focus on depression and anxiety), warning signs of mental disorders, crisis and suicidal risk and its management, importance of mental health issues in the workplace, and communication strategies for supporting employees with mental health problems.

As a general result we found a significant reduction in stigmatising attitudes in almost all studies (20/22), using 10 different instruments/scales. A detailed overview of study characteristics is presented in online supplemental table 1 and the main findings of each study are presented in online supplemental table 2.

### Mode of delivery
In the next section we will shortly describe some main features of the 22 studies. First, we present the online interventions, then the face-to-face interventions and finally the blended interventions. Within each category we begin with presenting studies with an RCT design followed by studies with a quasi-experimental design or other study designs.

### Online interventions
Four out of the 22 studies delivered the intervention in an online format.[28–31] Out of the four studies, three found significant positive effects on stigmatising attitudes, while one intervention did not find a positive effect after the intervention.[31] The average length of these online interventions was 146 min, the shortest being 30–45 min and the longest 6 hours. The positive effects were maintained at 3 months[29] and 6 months follow-up.[28 30]

### RCT design studies
Griffiths *et al* investigated the effectiveness of a 1 hour long online mental health programme for employees of governmental organisations (n=507).[28] Significant reduction measured by the personal subscales of The Depression and Generalised Anxiety Stigma Scales[32 33] was found postintervention and 6 months follow-up. Shann *et al* delivered an online leadership intervention (n=311).[30] Even a short, 30–45 mins duration intervention resulted in a significant reduction in stigma scores even at 6 months follow-up, which was measured by a 12-item Managerial Stigma Towards Employee Depression Scale.[34]

### Studies with non-RCT design
Paterson *et al* delivered a 6 hours long online workplace intervention (n=134).[31] No significant difference in premeasures and postmeasures stigma scores between intervention and control group was found, and the

methodological quality was rated as weak. The adopted version of King's Stigma Scale was used.[35] Hanisch *et al* delivered a 2-hour digital training for managers (n=48).[29] The intervention resulted in significant reduction regarding stigmatisation towards people with mental health problems, but no control group was enrolled. The Opening Minds Scale for Workplace Attitudes was used post-training and at 3 months follow-up.[35]

### Face-to-face interventions
Most of the studies used a face-to-face approach (15 out of 22). The average length of these interventions was 10.1 hours (=606 min), the shortest being 2 hours and the longest 16 hours. Only one intervention did not find a significant positive effect on stigmatising attitudes,[36] and one revealed rebound effect at 3 months follow-up.[37] Two further studies did not have a follow-up measurement.[38 39] The length of the follow-up varied between 1 month to 2 years.

### Studies with RCT design
Six studies used RCT designs, one rated as a methodologically strong study: Svensson and Hansson[40] conducted a 12-hour long training for public sector employees (n=199). A vignette version of the Depression Personal and Perceived Stigma Scale[32] showed significant reduction in personal stigma towards people with depression after 6 months and even at 2 years follow-up, but no significant changes were found in the control group. Similarly, the other four studies[37 41–43] found significant reduction in stigmatising attitudes in their intervention group post-training, and 1–3 months follow-up, but no significant changes were found in the control groups. The effects of 3–7.5 hours face-to-face trainings were measured by the modified version of the Depression Stigma Personal Subscale,[32] the Opening Minds Scale for Workplace Attitude,[35] the Opening Minds Stigma Scale for Healthcare Providers[44] and the Mental Health Knowledge Scale.[43] Fire service line managers (n=106) were randomly assigned to either a 2 days or a 12 hours long training group or a control group (1 hour leaflet session).[39] The locally developed Knowledge and Efficacy about Mental Health Problems Scale revealed statistically significant improvements in stigma on mental health pretraining and post-training in both training settings, but not in the control group.

### Studies with non-RCT design
Five out of six *weak methodological quality* interventions were performed using a face-to face non-RCT design.

Bond *et al*[45] delivered a 4-hour course for employees in support services (n=284). Significant reduction was found on stigmatising items measured by an adapted version of the Depression Stigma Scale[12] after the course and 6 months follow-up. Kubo *et al*[46] delivered a 2 hours long education programme (n=91). Right after the intervention, the Japanese version of the Links Perceived Devaluation-Discrimination Scale[10] showed a significant decrease in negative attitudes towards mental health problems, but this difference was not maintained after 1 month. Although there

was a long-term (2 years) effect in perceived mental health stigma in Kristman's *et al's*[47] 2 years long quasi-experimental study (n=89), the methodological quality of the study was assessed as weak. Quinn *et al*[48] conducted a 6 hours long training course for telecommunication workers (n=101). Relevant questions gathered from the Scottish Public Attitudes Survey[49] revealed a significant decrease in stigmatising attitudes between preintervention and postintervention, however the methodology was rated as weak. Stelnicki *et al*[50] conducted a 16 hours long programme for public safety personel (n=136) which resulted in significant decrease in stigma scores measured by the Opening Minds Scale for Workplace Attitude.[35]

### Five other face-to-face studies were rated as having moderate methodological quality

Dobson *et al*[51] (n=1292) and Szető *et al* (n=5598) investigated the effects of a 4 hours and 8 hours long stigma reduction programme for front-line workers and managers.[52] In both studies, the Opening Minds Scale for Workplace Attitudes[35] showed a significant reduction in stigma for the total scale and all the subscales between preintervention and postintervention and 3 months follow-up in both groups. In their longitudinal cohort study, Hamann *et al*[38] delivered a 1–1.5 day long face-to-face educational workshop for leaders and human resources department employees (n=580). Postintervention, the Depression Stigma Personal Subscale[32] showed a significant decrease, but no follow-up measure was performed. On the other hand, reduction in stigma was not significant in a 1 hour training followed by a 4 hours gatekeeper training for Australian Mates in Mining co-workers (n=1275) and 117 supervisors.[36] Mental health stigma was measured by the Perceived Stigma Scale.[53]

### Blended studies

All of the blended design studies used randomised designs. In a study by Moll *et al* with strong methodological quality, mental health literacy training was delivered to healthcare workers (n=192) in either face-to-face or blended setting.[54] Both interventions resulted in a significant reduction of stigmatising beliefs, but a longer effect was seen by the blended intervention at 6 months follow-up, which was measured by the Opening Minds Scale for Healthcare Providers.[44] In a study by Reavley *et al* 608 public sector employees were randomised into different interventions: two MHFA (Mental Health First Aid) and PFA (Psychological First Aid) online courses and a blended MHFA one.[55 56] Significant reduction in stigma scores were found in each intervention groups post training and 1 year follow-up and the Personal Stigma Scale[12] showed no significant difference between online and blended courses. Lam *et al's*[57] 3 months long study delivered an online Mental Health First Aid training combined with face-to-face sessions for various large enterprise employers (n=456). The strong methodological quality study resulted in a significant reduction of stigma scores post-training and at 3 months follow-up.

## DISCUSSION

The main aim of this systematic review was to identify and evaluate the effectiveness of different workplace-based antistigma interventions, focusing on reducing stigmatising attitudes and discrimination of people with mental illness. The review included interventions that were delivered to employees and employers. A specific focus was placed on SMEs.

Twenty-two articles met the inclusion criteria and we found an overall positive effect for most of the interventions irrespective of the mode of delivery. Three of the four studies using online interventions found positive effects. Among the 15 face-to-face interventions, only 1 study did not find an effect, although a few studies only found short-term effects. This finding appears to indicate that online antistigma interventions can be just as effective as face-to-face interventions. Similarly, a study comparing training for managers to improve their confidence in supporting the mental health of their employees found both the online and face-to-face versions to be effective.[23] As for the intensity of the intervention, we can conclude that the average length of online interventions was substantially shorter compared with those delivered face to face (146 min vs 606 min on average).

The finding that online interventions might be just as effective as face-to-face interventions was also confirmed by two further randomised controlled studies identified in this review. Reavley *et al*[55 56] found no significant difference between the effectiveness of blended and purely online interventions on stigmatising attitudes, and a longer-lasting positive effect was found in a blended intervention compared with its face-to-face version in another study.[54] These results underline the possible benefits of online interventions over the conventional face-to-face approaches: online interventions are shorter, need no presence of the professionals/trainers, and they have particular potential for the workplace as they can be tailored to participant or workplace needs (ie, can be used anytime during the day), which may also have favourable cost implications. These features make them especially attractive for SMEs as they typically have fewer resources for implementing workplace mental health interventions. Online interventions can also be beneficial during public health emergencies (such as the COVID-19 pandemic) when face-to-face contact is reduced or not possible.

We can conclude that the quality of the interventions has improved since Hanisch *et al's* review,[24] having only three overlapping studies with this previous review.[39 40 47] We identified studies with larger sample size and longer-lasting effects. Our review also confirms the findings of the previous review with more studies with higher methodological quality. However, in this review the majority of the identified studies did not have a control group and the dropout rate in some studies was high. Only 2 of the 22 studies were rated to have strong methodological quality. The majority of the programmes used a multitude of intervention techniques targeting both employees and leaders, which may have made the intervention more

effective, but this produces difficulties in terms of identifying the most effective elements for stigma reduction.

With regard to evaluation aspects, 17 studies included follow-up measurements after the intervention, with the duration varying from 1 month to 2 years. Most of the studies used a 1–6 months follow-up, only two programmes followed their participants for 2 years, and both found that the effects were maintained. A few studies however, reported only short-term effects. It remains unclear why some interventions demonstrate long-term effects while other studies only achieved short-term effects. More studies with longer follow-up time and more studies with more details about the content of the intervention are needed to investigate this further.

Despite the overall positive outcomes on stigmatising attitudes by the reviewed studies, it would be important to know if employees actually experience a reduction in exposure to mental health related stigma from their colleagues and managers following the interventions. Measurement tools assess changes in attitudes that do not always translate into differences in behaviour and other measures should more frequently be applied in these studies, such as the willingness to seek or offer help.

One of the two studies, which did not find a significant reduction in stigmatising attitudes after the intervention, investigated the effects of a 6 hours long online training programme.[31] Authors concluded that the stigma questionnaire[33] used in their evaluation may not have been sensitive enough to capture improvement in mental health related stigma in the workplace context. Similarly, a non-validated stigma-measuring scale could be the reason of another intervention which seems to have no significant reduction in stigma scores.[36]

Although our primary aim was to review changes in mental health related stigma, other results are also noteworthy. For example, some interventions were also found to contribute to increased mental health literacy[41 54] and intention to seek help.[28] Increased resilience[51 52] and help-seeking behaviour[28 54] were also observed, confirming previous findings by Hanisch et al.[24]

Workplace-based mental health stigma reduction programmes appear to have very similar key objectives and approaches, although we noted a tendency to use different evaluation approaches using different scales. The use of appropriate, psychometrically sound scales to assess stigma is crucial and facilitates comparison of findings. Both of the interventions[31 36] with no significant reductions in stigma scores applied scales that may not have been sensitive enough in workplace settings. Moreover, some researchers used semistructured interviews or primarily qualitative methods for evaluating programme effectiveness meaning they were excluded from our review, although these also found a reduction in participants' stigmatising attitudes.[58]

In sum, our main objective was to review effective workplace-based interventions for addressing mental health related stigma with a particular focus on SMEs. Unfortunately, our results did not entirely meet our expectations, as none of the reviewed interventions targeted SMEs specifically. Possible reasons behind this may be due to data protection reasons as limited data on the exact size and type of the organisations were noted. Most of the interventions were conducted in larger companies or public organisations, and therefore it is difficult to determine their feasibility in smaller enterprises with smaller numbers of employees and supervisors. However, we identified positive effects in studies where differently sized companies participated. Stigma reduction in SME workplaces therefore remains unaddressed, although our review did add some new perspectives for smaller enterprises.

Our purpose to review interventions with appropriate methodology has produced rather positive results. The reviewed papers indicate that the included interventions produced for the most part significant reductions in stigmatising attitudes for both employees and managers, and despite variation in methodology, common conclusions could be drawn.

## Limitations

Notwithstanding the positive results of this review, several limitations should be mentioned. Only English language articles were included from five electronic databases, but we did not use occupational health databases for primary literature.

We have identified a clear dominance of interventions targeting higher educated white-collar employers and employees, inhibiting the generalisability of effectiveness to less educated or blue-collar employees. In addition, all studies were conducted in either European countries, North-America, Australia or Japan, therefore not representing experiences from other parts of the world, with larger parts of the populations with lower economic status. Only studies with quantitative measurement were included in this review, however studies with interview or focus group designs could provide important additional information. Similarly, we did exclude studies with no direct measure on stigma, however attitudes towards mentally ill patients and knowledge of mental health are important factors of stigmatising behaviour. Given the diverse study designs and outcome measures, it was not possible to conduct a meta-analysis.

Having based our review on quantitative studies we found that most programmes were effective in changing stigmatising attitudes and in some studies also were able to lead to behaviour change. However, this review does not provide a better understanding of the mechanisms that lead to these changes. The knowledge about the effectiveness of the antistigma interventions presented in this review therfore should be supplemented with other reviews, including more or only qualitative studies, to investigate these aspects. Another important aspect of future studies can be the evaluation of which elements of interventions act on the level of individual and structural stigma separately. Again this also requires studies based on qualitative methodology.

## CONCLUSIONS

A large proportion of the workforce could benefit from workplace-based interventions aimed at reducing mental health related stigma. Although we did not find interventions focusing specifically on SMEs, we can derive important findings from our review. Online antistigma interventions could have several benefits for smaller enterprises; they are shorter, and appear to have the same positive effects on stigmatising attitudes as face-to-face interventions. These could be very important factors for professionals when trying to choose an intervention for their company.

Furthermore, investigations of the feasibility of these programmes in smaller enterprises with less resources are needed, and more studies should go beyond measuring only attitudes.

### Author affiliations

[1]Institute of Behavioural Sciences, Semmelweis University, Budapest, Hungary
[2]School of Public Health, University College Cork, Cork, Ireland
[3]National Suicide Research Foundation, Cork, Ireland
[4]National Research Centre for the Working Environment, Copenhagen, Denmark
[5]Centro Fórum Research Unit, Institute of Neuropsychiatry and Addictions (INAD), Hospital del Mar Medical Research Institute (IMIM), Parc de Salut Mar, Barcelona, Centro de Investigación Biomédica en Red de Salud Mental (CIBERSAM), Barcelona, Spain
[6]University Pompeu Fabra, Barcelona, Spain
[7]Finnish Institute for Health and Welfare (THL), Helsinki, Finland
[8]German Depression Foundation, Leipzig, Germany
[9]Department for Psychiatry, Psychosomatics and Psychotherapy, Depression Research Center of the German Depression Foundation, Goethe University, Frankfurt, Germany
[10]Phrenos Center of Expertise for Severe Mental Illness, Utrecht, The Netherlands
[11]Altrecht Mental Health Care, Utrecht, The Netherlands
[12]Mental Health Center, Prizren, Kosovo
[13]Department of Public Health, University of Medicine, Tirana, Albania
[14]LUCAS, Centre for Care Research and Consultancy, KU Leuven, Leuven, Belgium
[15]Australian Institute for Suicide Research and Prevention, School of Applied Psychology, Griffith University, Brisbane, Queensland, Australia
[16]Faculty of Epidemiology and Population Health, London School of Hygiene and Tropical Medicine, London, UK

**Collaborators** MENTUPP Consortium.

**Contributors** MDT, GP, SI and EA conceived the idea for the study. MDT, GP, SI, BA and CL planned the study design. MDT, SI and GP carried out the search and quality assessments with inputs from BA and CL. MDT and GP wrote the first draft of the report with inputs from BA, JCS, HR, GC, SS, NF, GQ, FT, VR, SM and ACP. All authors contributed to the interpretation of findings and critical revision of the manuscript. All authors approved the final version of the manuscript for submission. MDT submitted the manuscript and she is responsible for the overall content as guarantor.

**Funding** This work was supported by the European Union's Horizon 2020 Research and Innovation Programme grant number 848137.

**Competing interests** None declared.

**Patient and public involvement** Patients and/or the public were not involved in the design, or conduct, or reporting, or dissemination plans of this research.

**Patient consent for publication** Not applicable.

**Provenance and peer review** Not commissioned; externally peer reviewed.

**Data availability statement** No data are available.

### ORCID iDs

Mónika Ditta Tóth http://orcid.org/0000-0001-8903-4802
Caleb Leduc http://orcid.org/0000-0003-1140-0184
Birgit Aust http://orcid.org/0000-0002-4184-6577
Benedikt L Amann http://orcid.org/0000-0002-4407-1519
Johanna Cresswell-Smith http://orcid.org/0000-0003-2740-3830
Hanna Reich http://orcid.org/0000-0002-9577-1144
Grace Cully http://orcid.org/0000-0002-9236-1545
Sarita Sanches http://orcid.org/0000-0001-6675-927X
Naim Fanaj http://orcid.org/0000-0001-7375-643X
Fotini Tsantila http://orcid.org/0000-0002-1229-6318
Victoria Ross http://orcid.org/0000-0002-6768-1205
Sharna Mathieu http://orcid.org/0000-0003-0784-0638
Arlinda Cerga Pashoja http://orcid.org/0000-0002-7029-947X
Ella Arensman http://orcid.org/0000-0003-0376-1203
György Purebl http://orcid.org/0000-0002-9750-2001

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
