## [Reviewer comments · BMJ Open]

ARTICLE DETAILS

TITLE (PROVISIONAL)	Evidence for the effectiveness of interventions to reduce mental health related stigma in the workplace: A Systematic Review
AUTHORS	Ditta Tóth, Mónika; Ihionvien, Sarah; Leduc, Caleb; Aust, Birgit; Amann, Benedikt; Cresswell-Smith, Johanna; Reich, Hanna; Cully, Grace; Sanches, Sarita; Fanaj, Naim; Qirjako, Gentiana; Tsantila, Fotini; Ross, Victoria; Mathieu, Sharna; Pashoja, Arlinda; Arensman, Ella; Purebl, György

VERSION 1 – REVIEW

REVIEWER	Moll, Sandra McMaster Univ, School of Rehabilitation Science
REVIEW RETURNED	22-Sep-2022

GENERAL COMMENTS	Overall, this paper was well written with a clear focus on exploring interventions to reduce mental health related stigma in the workplace. The review was conducted in a rigorous, systematic way, with clear reporting of findings. My primary concern was under-theorizing of the central construct of stigma and stigma-related interventions. If examining effectiveness of workplace interventions to address stigma, more attention is needed to unpacking the nature of interventions themselves, including relationship between knowledge, beliefs and behaviours, and the mechanisms of behaviour change. Were ‘best practices’ of stigma reduction, for example, incorporated into the interventions? It would be important to note the number of studies which incorporated standardized approaches such as MHFA or the Working Mind versus other non-standardized mental health awareness approaches. Critical reflection on contact-based education (considered a ‘best practice’), and approaches which target individual versus structural stigma within workplaces might lead to a more robust scholarly contribution to the literature than simply noting whether the intervention was conducted online or in-person. There were a number of concerning inconsistencies in the paper that require clarification. The specified focus of the study was on SME’s, yet this was not part of the study inclusion/exclusion criteria. When reporting on sector and size of organization (page 8), the authors initially say that 3 studies focused on private sector organizations, then in the next paragraph report 4 studies in private sector. Supplementary table 1 is incomplete in terms of reporting on study designs: several were missing. Also, the authors note in the paper that there were 5 studies that used RCT designs, but Table 1 reflects 7 RCTs. Supplementary table 2 incorrectly classifies several of the studies as RCTs versus quasi-experimental studies. These are important issues that raise concerns about the reported findings.
---

	The end date of literature search was July 2021 - an updated search to Aug 2022 would ensure that the findings are current. Overall, I feel that the paper has merit, particularly given the importance of addressing workplace mental health, but revisions are needed to ensure that the paper reflects a credible contribution to the literature.
--	--

REVIEWER	Hicks, R. E. Bond University
REVIEW RETURNED	25-Sep-2022

GENERAL COMMENTS	This article meets a unique need and will interest many practicing professionals involved in stigma interventions. Your work as authors shows professional workmanlike adherence to the PRISMA standards at a high level, and your writing is clear and understandable. I would have liked more on the content of the various interventions but this was not part of your aim and I found sufficient references to this in the major Table on overview of study/article characteristics. . On some minor aspects- I would like to see more consistency in the tables and supplementary tables where statistics is reported- e.g., all decimals given as either the comma or the full-stop but not both; and the probability should be consistently given as p (in italics).. More specifically (including some questions)  1. p6 line 2: employees', not employee's 2. In the Study characteristics section, p8, I note no US studies appea. I thought this surprising. Any comment? (None needed for the paper but I wondered why). 3. I was also disappointed to see there were few experimental SME studies- but again, this is why you carried out the study. To see what the situation is. I appreciate your effort! That's it. A solid piece of work.
---

VERSION 1 – AUTHOR RESPONSE

Reviewer: 1

Dr. Sandra Moll, McMaster Univ
Comments to the Author:

Overall, this paper was well written with a clear focus on exploring interventions to reduce mental health related stigma in the workplace. The review was conducted in a rigorous, systematic way, with clear reporting of findings.

Thank you, we really appreciate your opinion.

My primary concern was under-theorizing of the central construct of stigma and stigma-related interventions. If examining effectiveness of workplace interventions to address stigma, more attention is needed to unpacking the nature of interventions themselves, including relationship between knowledge, beliefs and behaviours, and the mechanisms of behaviour change. *Were ‘best practices’ of stigma reduction, for example, incorporated into the interventions?*

Thank you very much; this is a very important concern. It would be crucial to unpack which active elements of the programs were responsible for the changes that were found. Because of our search strategy and inclusion criteria we cannot distinguish between different types of outcomes (knowledge, beliefs and behaviours), but only can report about outcomes measured with direct stigma measuring instruments, which mostly measure attitudes. Because of that, we also can not investigate the aspects that you are asking for.

As the outcome measures can not provide sufficient evidences about the key points of the change, the only conclusion we can draw is the necessity of the targeted investigation of the mechanism of change in further studies, which require a different type of studies using other methodology (i.e. interpretative phenomenological analysis).

We are aware of that we in our review only looked for studies that for the most part measured attitudes and that changes in attitudes not necessarily translate into changed behaviour. Nevertheless, we consider changes in attitude a first step on the way to behaviour change and wanted to investigate how effective programs are in achieving that outcome. Nevertheless, we added a few more self-critical sentences in the manuscript and thereby make clear that despite the positive results found in this review more needs to be done to achieve behaviour change.

See changes in Limitations section accordingly.

It would be important to note the number of studies which incorporated standardized approaches such as MHFA or the Working Mind versus other non-standardized mental health awareness approaches.

Thank you for your comment. We added the number of the studies using standardized vs non-standardized approaches to the 'Interventions' section.

Critical reflection on contact-based education (considered a 'best practice'), and approaches which target individual versus structural stigma within workplaces might lead to a more robust scholarly contribution to the literature than simply noting whether the intervention was conducted online or in-person.

Thank you very much for this comment, this is indeed a crucial point of any stigma-reduction efforts. However, in this review, our focus was on interventions that can be used in SMEs and therefore we were particularly interested in interventions that would also be possible to implement SMEs that typically have fewer resources for implementing workplace mental health interventions as pointed out in the introduction and in the discussion part of the manuscript. Nevertheless, we agree that more research needs to be done to better understand the mechanisms and specific advantages of contact-based education and believe that more qualitative studies are needed for this. We have added a paragraph under Strength and Limitations to address this issue.

There were a number of concerning inconsistencies in the paper that require clarification.

Thank you for this valuable remark. We corrected each of your points.

The specified focus of the study was on SME's, yet this was not part of the study inclusion/exclusion criteria.

Thank you for this comment. We searched broadly, including all types of organization sizes, but also specifically for small and medium size enterprises as can be seen in our search string (see Appendix 1). However, we did not use size of the organization as an inclusion criteria, but included all anti-stigma intervention studies that fulfilled the inclusion criteria regardless of size of the organization. Nevertheless, we investigated size of organization in each of the identified studies and documented it as part of the data extraction (see data extraction paragraph in the manuscript and supplementary Table 1). Although we did not find any studies that exclusively focussed on SMEs this procedure made it possible to identify a few studies that also include SMEs.

When reporting on sector and size of organization (page 8), the authors initially say that 3 studies focused on private sector organizations, then in the next paragraph report 4 studies in private sector.

Thank you for making us aware of this inconsistency, but due to the updated search now so it is "4 studies"

Supplementary table 1 is incomplete in terms of reporting on study designs: several were missing.

Thank you for making us aware of this mistake. We added the missing study designs.

Also, the authors note in the paper that there were 5 studies that used RCT designs, but Table 1 reflects 7 RCTs.

Thank you for your comment; we corrected it in the text. However, due to the updated search the review now includes 9 interventions with an RCT design.

Supplementary table 2 incorrectly classifies several of the studies as RCTs versus quasi-experimental studies. These are important issues that raise concerns about the reported findings.

Thank you for your comment, we corrected Table 2 accordingly. However, due to the updated search the new numbers are now: 9 RCT design and 13 non-RCT design studies.

The end date of literature search was July 2021 - an updated search to Aug 2022 would ensure that the findings are current.

Thank you for your suggestion. We updated our search to November 2022 and found 4 more relevant articles from the previous year. Three articles presented results from anti-stigma interventions, while one article (Reavley et al. 2021) presented long-term effects from an intervention study that we had previously identified (Reavley et al., 2018). Therefore, the review is now based on 23 articles that report about 22 intervention studies. Please, see our changes in the main document and Table 1, 2, 3 and 4.

Overall, I feel that the paper has merit, particularly given the importance of addressing workplace mental health, but revisions are needed to ensure that the paper reflects a credible contribution to the literature.

Thank you for your positive feedback and your comprehensive review.

Reviewer: 2

R. E. Hicks, Bond University
Comments to the Author:

This article meets a unique need and will interest many practicing professionals involved in stigma interventions. Your work as authors shows professional workmanlike adherence to the PRISMA standards at a high level, and your writing is clear and understandable.

Thank you for your positive feedback, it is highly appreciated.

I would have liked more on the content of the various interventions but this was not part of your aim and I found sufficient references to this in the major Table on overview of study/article characteristics.

Yes, we agree with your comment. It would be indeed very interesting to write more about the specific interventions, unfortunately, due to the space limitation there was no possibility to provide more details about the content of the various interventions in the text.

On some minor aspects- I would like to see more consistency in the tables and supplementary tables where statistics is reported- e.g., all decimals given as either the comma or the full-stop but not both; and the probability should be consistently given as p (in italics)..

Thank you for your important note, we corrected Table 2 accordingly.

More specifically (including some questions)

1. p6 line 2: employees', not employee's – corrected

2. In the Study characteristics section, p8, I note no US studies appear. I thought this surprising. Any comment? (None needed for the paper but I wondered why).

This was a surprise for us as well. However, there are 7 Canadian interventions, none in the US. We also checked the review study by Hanisch et al. from 2016 that identified 16 studies published between 2004 and 2014. Also they only identified one study from the US. Therefore, it seems like not many workplace based anti-stigma intervention studies are being conducted in the US.

3. I was also disappointed to see there were few experimental SME studies- but again, this is why you carried out the study. To see what the situation is. I appreciate your effort!

Thank you for your comment! The lack of interventions in SMEs was a surprise for us as well, we also did not expect a total lack of studies focusing exclusively on SMEs. Our project 'Mentupp' is aiming to fill this gap with regard to the lack of mental health related interventions in Small and Medium sized enterprises.

That's it. A solid piece of work.

Thank you very much for your positive feedback.